# Research on Chinese Named Entity Recognition Based on BERT-CNN-BiLSTM-CRF Model with Fusion Multi-head Attention Mechanism

1st Qiu.Long.Li
*College of Electronic and Electrical Engineering*
Shanghai University of Engineering Science
Shanghai,China
2238983010@qq.com

2nd Shu.Jia.Yan
*College of Electronic and Electrical Engineering*
Shanghai University of Engineering Science
Shanghai,China
yanshujia@sues.edu.cn

3rd Qiang.Chen
*College of Electronic and Electrical Engineering*
Shanghai University of Engineering Science
Shanghai,China
sues_chen@sues.edu.cn

4th Kai.Zhang
*College of Electronic and Electrical Engineering*
Shanghai University of Engineering Science
Shanghai,China
775961503@qq.com

*Abstract*—In the domain of natural language processing, Named Entity Recognition (NER) has emerged as a pivotal task for knowledge graph construction and information extraction. Due to the characteristics of polysemy and ambiguous word boundaries, the identification and extraction of Chinese entities pose greater challenges. In response to the shortcomings of traditional language processing models in extracting local semantic features of Chinese entities and effectively obtaining information from different positions in text statements, this paper integrates the multi-head attention mechanism with BERT, CNN, BiLSTM, and CRF components to propose a new entity recognition model. The BERT pre-trainedmodel generates word vectors that encapsulate contextual semantic information. A Convolutional Neural Network (CNN) then combines and extracts multi-level semantic information to enrich the feature set. The Bidirectional Long Short-Term Memory Network (BiLSTM) further processes these vectors to extract features, while the multi-head attention mechanism enhances the model's capability to capture long-distance dependencies within the text. Finally, a Conditional Random Field (CRF) decodes and generates the sequence of entity labels. Experimental results demonstrate that the performance of this model outperforms other language processing models on both the low-sample Dgre dataset and the high-sample Duie dataset, showing more excellent performance on the latter.

*Keywords—Named Entity Recognition,BERT, CNN, BiLSTM, CRF,Multi-head Attention Mechanism*

## I. INTRODUCTION

Named entity recognition (NER) is an important research direction in the field of natural language processing (NLP), aiming to identify and extract various types of named entities from unstructured Chinese text, such as book titles, people, etc. For example, in the sentence" 'Journey to the West' is written by Wu Chengen and is a novel," the entities include the personal entity' Journey to the West.' and the book title entity "Wu Chengen." NER research is rapidly developing both domestically and internationally, and it has a wide range of applications in various NLP tasks such as knowledge graph construction, machine translation, and knowledge base construction.

In recent years, deep learning methods have been widely used in named entity recognition (NER). J. Hammerton was the first to apply Long Short-Term Memory networks (LSTM) to the study of text entity recognition, making the LSTM-CRF architecture the foundational structure for entity recognition[1]. Gao Xiang and others proposed using a model that combines Conditional Random Fields (CRF) with Long Short-Term Memory neural networks (LSTMs), known as the LSTM-CRF model. By incorporating pre-trained word embedding vectors and different word position tagging sets, they performed entity recognition on military movement texts.Lample and Guillaume et al.[2], building upon the LSTM-CRF model, proposed models that combine Bidirectional Long Short-Term Memory networks (Bi-LSTM) with CRF[3-5]. This structure is capable of capturing bidirectional semantic information in text, showing excellent performance in text named entity recognition tasks, achieving an F1 score of 90.94% on the CoNLL-2003 dataset[6]. These methods have improved the accuracy and efficiency of text entity recognition, but they all have a drawback: they cannot handle the issue of word polysemy. Polysemy is a characteristic of Chinese text. This means that these methods can only ignore the contextual semantic information of words and only process independent word feature vectors, leading to the problem of not achieving high accuracy in Chinese entity recognition. The BERT model proposed by the Google team in 2018[7]utilizes a bidirectional Transformer neural network as an encoder, effectively addressing the issue of polysemy. Shen Tongping and colleagues introduced a Chinese named entity recognition model based on BERT+BiLSTM+CRF[8] which can effectively identify and extract entities, achieving a recognition accuracy of 95.91% in the People's Daily corpus. While these methods effectively address the issue of polysemy in Chinese text, they still face the problem of being inefficient in extracting semantic features from local areas within Chinese entities.

In response to these issues, this paper proposes a Chinese named entity recognition model that integrates BERT, CNN, BiLSTM, and CRF. This model facilitates the efficient extraction of semantic features from localized areas within Chinese entities. Furthermore, to resolve the problem of inefficient information retrieval from various positions in text sentences, this paper integrates a multi-head attention mechanism. This mechanism is designed to acquire long-distance dependencies from Chinese text and to weight the feature vectors output by the subsequent layers, thereby significantly improving the model's precision and recall rates.

## II. Model Introduction

### A. Model Summary

The BERT-CNN-BiLSTM-CRF model, enhanced with a multi-head attention mechanism as presented in this paper, is illustrated in Fig. 1 and comprises five primary layers. The pre-trained BERT layer captures the intricate relationships between vocabulary in the input text sequence, generating embedded representations for each token. These representations serve as input to the CNN layer, which is responsible for acquiring local features. The convolutional features are then fed into the BiLSTM layer for bidirectional training, further extracting features from the text. The BiLSTM's output is passed to the attention layer, where the multi-head attention mechanism is utilized to augment the model's capacity to capture information across various subspaces, leading to richer feature representations. This attention layer also strengthens the model's ability to handle long-distance dependencies, particularly in the context of lengthy texts. Ultimately, the CRF layer effectively models the dependencies between predicted tags, ensuring the acquisition of the correct globally optimal sequence.

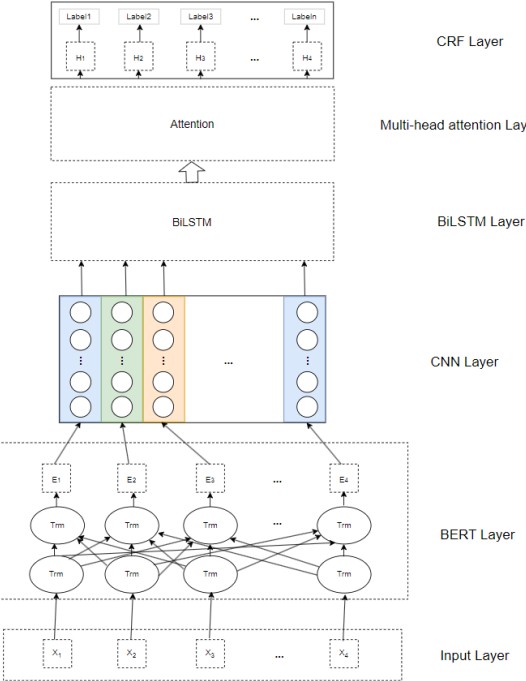

Fig. 1. The BERT-CNN-BiLSTM-CRF model framework integrated with the multi-head attention mechanism.

In addition to the aforementioned five layers, we have also added two linear layers to ensure the matching of input and output dimensions across the layers.

### B. BERT Model

In natural language processing, transforming textual information into corresponding word vector embeddings for incorporation into models is an important task[8]. In the Chinese language, the phenomenon of polysemy is very common, meaning that the same word can have different meanings in different contexts. Traditional language models, such as GloVe, Word2Vec, and GPT, cannot solve the problem of polysemy; the same word, regardless of its specific meaning within a sentence, is represented by the same vector. For example, in the phrases "开会"(hold a meeting) and "开门"(open the door), the word "开"(open) expresses completely different meanings, but in these models, the two instances of "开" are treated as having the same meaning, with identical word vector values.

The BERT model, introduced by Google's team in 2018, effectively tackles the challenge of word polysemy. It adopts a bidirectional Transformer architecture as its encoder, allowing it to consider contextual inputs when predicting subsequent characters. Beyond this, BERT trains deep bidirectional representations using a straightforward approach: it randomly masks 15% of the input tokens and then predicts these tokens solely based on the surrounding context. This technique is known as the "Masked Language Model" (MLM). The word vectors produced by the BERT model are composed of three components: token vectors, segment vectors, and positional vectors, as illustrated in Fig. 2.

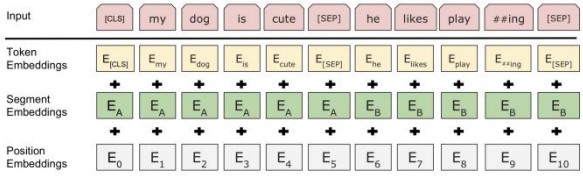

Fig. 2. BERT word vector

Among them, the special token [CLS] signifies the beginning of a text sequence, while the special token [SEP] denotes the separation between sentences or the end of a text sequence.

### C. CNN Module

Initially utilized mainly in the realms of image learning and computer vision, Convolutional Neural Networks (CNN) were integrated into Natural Language Processing (NLP) after Meek C[9] confirmed their efficacy in syntactic text analysis[10]. The first model of CNN, LeNet5, was introduced by Fukushima. Within NLP, CNN generally comprises an input layer, convolutional layers, pooling layers, and an output layer. The convolutional layers detect local features by moving filters

across the input data, and the pooling layers reduce the spatial dimensions of these features while enhancing the model's robustness to changes in the position of the features. This architecture allows CNN to effectively capture n-gram features within text, encoding the local semantic information of words.

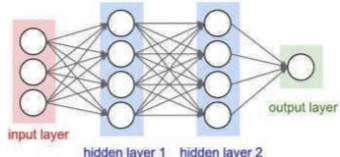

Fig. 3. convolutional neural network

In processing text, a unidirectional CNN can only capture local features within a specific region and cannot consider long-distance dependencies across regions. However, in natural language processing, text information often requires taking into account a broader context. For example, in the sentences "快快长大" (grow up quickly) and "开门" (open the door), the meaning of the individual words "开" (open) and "长" (grow) depends on the context of the entire phrase or sentence. Therefore, this paper adopts a bidirectional CNN.

Bidirectional CNNs apply convolutional operations in both the forward and reverse directions, allowing them to consider both preceding and following information within a text sequence. This enables a more comprehensive understanding of word semantics across various contexts. Such a model architecture is especially crucial for handling natural language texts that have intricate dependencies, as it can markedly improve the performance of tasks like named entity recognition. By integrating the CNN's ability to extract local features with other modules that capture long-distance dependencies, it is possible to build a more robust and powerful natural language processing system.

In order to match the feature dimensions output by the CNN with the input requirements of the BiLSTM, a linear layer has been incorporated after the CNN layer. This adjustment reduces the dimensionality of the CNN's output features from 256 to the 128 dimensions necessary for the LSTM layer.

### D. BILSTMModule

The LSTM (Long-Short Term Memory) network is a specialized type of Recurrent Neural Network (RNN). A unidirectional LSTM is limited to processing only the forward information within a text sequence and cannot handle backward information. This paper employs a BiLSTM module, which is an advanced recurrent neural network structure. It combines the LSTM's capacity for capturing long-term dependencies with the bidirectional processing's ability to perceive context. This architecture not only addresses the issues of gradient vanishing or explosion but also provides a holistic understanding of the sequence by maintaining two hidden states at each timestep to capture both the forward and backward information of the sequence. The BiLSTM structure is illustrated in Fig. 4.

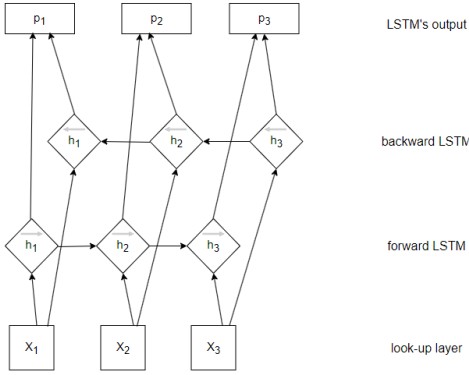

Fig. 4. BiLSTM Structure Diagram

### E. AttentionModule

The multi-head attention mechanism is akin to the human attention allocation when processing complex information. When we switch our focus between multiple tasks or information sources, we are able to concentrate on the most critical parts while disregarding secondary details. In this article, the multi-head attention mechanism is applied to the output of the BiLSTM layer to extract feature information that is crucial for entity recognition. This mechanism processes information in parallel through multiple "heads," each focusing on different aspects of data features, and then integrates these features to form a more useful feature representation for upper-layer tasks.

The core advantage of the multi-head attention mechanism lies in its ability to finely assign weights to the feature vectors output by the BiLSTM layer. It not only emphasizes features that are essential for entity recognition but also enhances the model's ability to focus on key information by disregarding irrelevant features. The application of this mechanism allows the model to directly assess which features are more important for specific downstream tasks by examining the weights of different heads, thereby optimizing the model's selective attention and improving overall recognition accuracy and model performance.

Assume we have a set of input sequences X with dimensions(m,$d_{model}$), where mm is the sequence length and $d_{model}$ is the model's dimension. In multi-head attention, we first linearly project Xonto three different matrices $W^Q$, $W^K$, and $W^V$ to obtain the queries (Q), keys (K), and values (V):

$$Q = XW^Q \tag{1}$$

$$K = XW^K \tag{2}$$

$$V = XW^V \tag{3}$$

Here, $W^Q, W^K, W^V$ are learnable weight matrices that map the input X from dimension $d_{model}$ to $d_k$ and $d_v$, where $d_k = d_v = \dfrac{d_{model}}{h}$, and h is the number of attention heads.

Next, for each head i, the attention weights $Attention_i(Q,K,V)$ are computed using the standard scaled dot-product attention formula:

$$Attention_i(Q,K,V) = soft\max(\frac{Q_i K_i^T}{\sqrt{d_k}})V_i \tag{4}$$

In this formula, $Q_i$, $K_i$ and $V_i$ are the slices corresponding to the i-th head from Q, K and V respectively. The softmax function is used to normalize the weights so that the sum of all weights is 1.

Finally, the outputs of all heads are concatenated and projected through another linear layer $W^O$ to obtain the final output:

$$MultiHead(Q,K,V) = W^o[\backslash concat(Attention_1,...,Attention_h)] \tag{5}$$

The '\concat' operation denotes concatenation along the feature dimension.

This paper has set up 8 attention heads in the attention layer for computation. Based on previous experiments, 8 heads have provided the best performance for the model on the given task.

In addition, we added a Linear fully connected layer after the attention layer to map the output of the attention layer to the desired label space.

### F. CRFModule

Conditional Random Field (CRF) is a statistical modeling tool used for sequence prediction tasks. It takes into account the inter-dependencies among labels within a sequence, making it well-suited for scenarios involving the labeling or classification of sequential data, such as Named Entity Recognition (NER) and Part-of-Speech (POS) tagging.

During the entity recognition process in Chinese text, the multi-head attention mechanism cannot account for the dependency relationships between labels, for instance, the 'I-ORG' tag should not directly follow the 'B-PER' tag. The CRF layer can effectively impose constraints on the dependencies between predicted labels, model the sequences of labels, and thus achieve the globally optimal sequence.

Mathematically, the CRF model can be expressed as follows:

$$P(y|x) = \frac{1}{Z(x)}\exp[\sum_{i,k} t_k(f_i(y_{i-1},y_i,x,i)) + \sum_{i,j} u_j(g_j(y_i,x,i))] \tag{6}$$

In the formula, P(y|x) represents the probability of the label sequence y given the input sequence x. $f_i$ and $g_j$ are feature functions, $t_k$ and $u_j$ are the corresponding weights, and Z(x) is

the normalization factor, ensuring the sum of probabilities for all label sequences equals 1.

The CRF model is trained by maximizing the log-likelihood function, which involves calculating the conditional probability of a label sequence y given a sentence X through equations (7) and (8). In this context, $y_x$ signifies all potential label sequences for the sentence X, and L refers to the specified loss function.

$$P(y|X) = \frac{\exp(score(X,y))}{\sum_{y' \in yX} \exp(X,y')} \tag{7}$$

$$L = \log(P(y|X)) \tag{8}$$

Here, y′ stands for the actual labels, while yX refers to the collection of all possible label sequences.

## III. EXPERIMENTAL RESULTS

### A. Experimental dataset

This study utilizes two datasets to assess the model's performance experimentally: ("Dgre")one with a small number of samples (a total of 907 instances) and ("Duie")another with a larger number of samples. The reason for evaluating the model's performance with the small-sample dataset is that named entity recognition tasks in low-resource settings have increasingly become a significant area of research in recent years.

The small-sample dataset is specific to the fault detection domain, comprising two types of entities: 'faulty equipment' (FE) and 'causes of failure'(COF). The large-sample dataset includes data from various fields, including book information and film and television sector and so on, encompassing a total of 24 entity types. This rich variety of entities is conducive to evaluating the model's ability to generalize.

### B. Dataset annotation and evaluation indicators

The BIO tagging scheme is a widely used sequence labeling approach, especially in the task of Named Entity Recognition (NER). This method involves assigning a tag to each word in the text to identify and categorize entities. The core of the BIO tagging method lies in its distinction between the start of an entity, the middle of an entity, and words that do not belong to any entity.

Specific annotation examples are shown in Table 1.

TABLE I.    ANNOTATED EXAMPLES

| Word | Label |
|------|-------|
| 汽 | B-FE |
| 车 | I-FE |
| 发 | I-FE |
| 动 | I-FE |
| 机 | I-FE |
| 进 | B-COF |
| 水 | I-COF |
| 了 | O |

The text evaluates the model's performance using Precision, Recall, and F1 score, with their respective calculation methods as follows:

$$P = \frac{E_P}{E_P + F_P} \times 100\% \qquad (9)$$

$$R = \frac{E_P}{E_P + F_n} \times 100\% \qquad (10)$$

$$F1 = \frac{2PR}{P + R} \times 100\% \qquad (11)$$

In the text, "$E_P$" refers to true positives, which is the number of correctly identified entities; "$F_P$" refers to false positives, which is the number of unrelated entities that were incorrectly identified; and "$F_n$" refers to false negatives, which is the number of related entities that were not recognized.

## C. Experimental environment parameters and result analysis

We conducted experiments on the proposed Chinese entity recognition model, which were based on the Pytorch framework. The specific experimental environment settings and The setting of hyperparameters in the modelare shown in Table 2 and Table 3.

TABLE II.    EXPERIMENTAL ENVIRONMENT SETUP

| Project | Environment |
|---|---|
| Operating system | Windows 10 |
| Memory | 24G |
| Graphics card | NVIDIA GeForce GTX 1080Ti 12G |
| Python version | 3.8.19 |
| Pytorch version | 1.8.0 |
| Transformer version | 4.27.4 |

TABLE III.    HYPERPARAMETER SETTING

| Parameter | Value |
|---|---|
| Handle maximum sequence length | 256 |
| Epoch | 50 |
| Dropout | 0.1 |
| CRF learning rate | 2e-3 |
| Adam epsilon | 1e-8 |
| Weight decay | 0.01 |

Table 4 reveals that under the condition of limited samples, the F1 scores of all models are not high. The BiLSTM-CRF model's F1 score is 2.11% higher than that of the BiLSTM, indicating that the CRF module can effectively improve the model's recognition effect. The CRF layer can effectively constrain the dependencies between predicted labels, model the label sequence, and thus obtain the globally optimal sequence. The BERT-BiLSTM-CRF model's F1 score is 2.74% higher than that of the BiLSTM-CRF. This is because the BERT model has strong semantic acquisition capabilities, which can

fully represent information about character-level, word-level, sentence-level, and inter-sentence relationships. After pre-processing with the BERT model, the trained word vectors can handle syntax and lexical information in different contexts, enhancing the model's generalization ability, and thereby improving the ability to identify and extract entities and relationships. The proposed model in this paper has an F1 score that is 2.24% higher than that of the BERT-BiLSTM-CRF, indicating that the CNN module and multi-head attention mechanism can effectively improve the model's recognition effect. The CNN layer can effectively enhance the model's ability to extract local features, and the multi-head attention mechanism can enhance the model's parallel computing and its ability to capture long-distance dependencies in the text.

TABLE IV.    "DGRE" DATASET TEST RESULTS  %

| Model | P | R | F1 |
|---|---|---|---|
| BILSTM | 66.67 | 70.08 | 69.87 |
| BILSTM-CRF | 69.39 | 70.59 | 71.98 |
| BERT-BILSTM-CRF | 72.46 | 76.56 | 74.72 |
| Model in this article | 74.58 | 76.61 | 76.96 |

From Table 5, it can be concluded that when there is an ample supply of samples, the performance advantage of the model proposed in this paper is better realized. The F1 score of the BiLSTM-CRF model is 2.14% higher than that of the BiLSTM, the F1 score of the BERT-BiLSTM-CRF model is 5.74% higher than that of the BiLSTM-CRF, and the F1 score of the model proposed in this paper is 5.08% higher than that of the BERT-BiLSTM-CRF.

TABLE V.    "DUIE" DATASET TEST RESULTS  %

| Model | P | R | F1 |
|---|---|---|---|
| BILSTM | 71.23 | 71.56 | 70.76 |
| BILSTM-CRF | 74.77 | 72.88 | 72.90 |
| BERT-BILSTM-CRF | 77.89 | 76.77 | 78.64 |
| Model in this article | 82.66 | 80.55 | 83.72 |

## D. Ablation experiment

To explore the impact of the CNN layer and the multi-head attention mechanism on model performance, the BERT-BiLSTM-CRF model is set as the baseline model. First, a CNN layer is added after the BERT layer to construct the BERT-CNN-BiLSTM-CRF model. Then, on the basis of the BERT-CNN-BiLSTM-CRF model, the multi-head attention mechanism is integrated to construct the BERT-CNN-BiLSTM-Att-CRF model. The experimental results of the above models on two datasets are shown in Tables 6 and 7.

TABLE VI.    "DGRE" DATASET TEST RESULTS  %

| Model | P | R | F1 |
|---|---|---|---|
| BERT-BILSTM-CRF | 72.46 | 76.56 | 74.72 |
| BERT-CNN-BILSTM-CRF | 74.02 | 75.98 | 76.45 |
| BERT-CNN-BILSTM-Att-CRF | 74.58 | 76.61 | 76.96 |

TABLE VII.    "DUIE" DATASET TEST RESULTS  %

| Model | P | R | F1 |
|---|---|---|---|
| BERT-BILSTM-CRF | 77.89 | 76.77 | 78.64 |
| BERT-CNN-BILSTM-CRF | 80.89 | 78.83 | 81.91 |
| BERT-CNN-BILSTM-Att-CRF | 82.66 | 80.55 | 83.72 |

The ablation study results in Tables 6 and 7 indicate that the BERT-CNN-BiLSTM-CRF model has an F1 score higher than the BERT-BiLSTM-CRF model on both datasets, which demonstrates that the CNN layer has an enhancing effect on the model's performance. From the precision, recall, and F1 scores achieved by the BERT-CNN-BiLSTM-Att-CRF model on both datasets, as shown in the tables, it can be seen that the BERT-CNN-BiLSTM-CRF model with the multi-head attention mechanism proposed in this paper performs well in the task of Chinese entity recognition.

## IV. CONCLUSION

In response to the shortcomings of traditional language processing models in effectively extracting semantic features of local areas within Chinese entities and acquiring information from different positions in text statements, a novel entity recognition model has been proposed.This model enhances the recognition performance in the task of Chinese named entity recognition by integrating the CNN layer and multi-head attention mechanism into the BERT-BiLSTM-CRF model. The next steps of work are, on one hand, to apply the model to specific domains to complete entity recognition and extraction, and on the other hand, to explore how the model can fully leverage its performance under the training environment with limited samples.

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
