# OpenReview forum: "Research on Chinese Named Entity Recognition Based on BERT-CNN-BiLSTM-CRF Model with Fusion Multi-head Attention Mechanism"
_IEEE.org/ICIST/2024/Conference — IEEE ICIST 2024 Conference Submission_

### Official Review · Reviewer_5mNL · 2024-08-22
**The topic of the paper is of good interest for the readers.**

**Rating:** 7
**Confidence:** 3

**Review:**

1.There are many grammatical and typographical errors in the manuscript. Please check the full text carefully and correct them. 2.The engineering background of the proposed problem should be stated more clearly to readers. The literature review is insufficient. Some recently published papers should be included in the references list. 3.Based on the proposed idea and obtained results in this paper, the authors should be able to present some more descriptions in conclusion part, for example, further study direction.

---

### Official Review · Reviewer_kXE8 · 2024-08-22
**This article is very interesting and a good one**

**Rating:** 7
**Confidence:** 3

**Review:**

This paper integrates the multi-head attention mechanism with BERT, CNN, BiLSTM, and CRF components to propose a new entity recognition model. The obtained result is valuable and can be accepted if the following problems can be clarified.
(1) In the introduction, the shortages of those relevant studies are suggested to be further summarized.
(2) In the end of Section 1, the organization of this study is suggested to be summarized.
(3) There exist several spelling and grammar errors. Please check carefully and further polish
(4) This paper utilizes Tables 1-7, however, the table captions inconsistently employ Roman numerals (I-VII), necessitating standardization to maintain uniformity.
(5) The references should be updated and their format standardized for enhanced consistency and accuracy.

---

### Official Review · Reviewer_NemQ · 2024-08-23
**accept**

**Rating:** 7
**Confidence:** 3

**Review:**

This paper integrates the multi-head attention mechanism with BERT, CNN, BiLSTM, and CRF components to propose a new entity recognition model.The theory is correct and can be accepted after responding the following comments.
(1)There are many typos and grammar errors. The authors should have a native English speaker or software packages to perform the editing check.
(2)In the introduction, it is not enough to state the current work. It should be expended and reconstructed.
(3)What is the contribution of the paper? It should be highlighted both in the introduction and in the content.

---

### Comment · Reviewer_kXE8 · 2024-08-21
**This article is very interesting and a good one**

This paper integrates the multi-head attention mechanism with BERT, CNN, BiLSTM, and CRF components to propose a new entity recognition model. The obtained result is valuable and can be accepted if the following problems can be clarified.
(1)	In the introduction, the shortages of those relevant studies are suggested to be further summarized.
(2)	In the end of Section 1, the organization of this study is suggested to be summarized.
(3)	There exist several spelling and grammar errors. Please check carefully and further polish
(4)	This paper utilizes Tables 1-7, however, the table captions inconsistently employ Roman numerals (I-VII), necessitating standardization to maintain uniformity.
(5)	The references should be updated and their format standardized for enhanced consistency and accuracy.

---

### Decision · Program_Chairs · 2024-09-06

Accept (Oral)